# Spatial-temporal dynamics of a microbial cooperative behavior resistant to cheating

Hilary Monaco [1,2,3✉], Kevin S. Liu [4,5], Tiago Sereno [2,6], Maxime Deforet [2,7], Bradford P. Taylor [2,8], Yanyan Chen [2], Caleb C. Reagor [1,9] & Joao B. Xavier [2✉]

Much of our understanding of bacterial behavior stems from studies in liquid culture. In nature, however, bacteria frequently live in densely packed spatially-structured communities. How does spatial structure affect bacterial cooperative behaviors? In this work, we examine rhamnolipid production—a cooperative and virulent behavior of *Pseudomonas aeruginosa*. Here we show that, in striking contrast to well-mixed liquid culture, rhamnolipid gene expression in spatially-structured colonies is strongly associated with colony specific growth rate, and is impacted by perturbation with diffusible quorum signals. To interpret these findings, we construct a data-driven statistical inference model which captures a length-scale of bacterial interaction that develops over time. Finally, we find that perturbation of *P. aeruginosa* swarms with quorum signals preserves the cooperating genotype in competition, rather than creating opportunities for cheaters. Overall, our data demonstrate that the complex response to spatial localization is key to preserving bacterial cooperative behaviors.

[1] Tri-Institutional PhD Program in Computational Biology and Medicine, New York, NY, USA. [2] Program for Computational and Systems Biology, Memorial Sloan-Kettering Cancer Center, New York, NY, USA. [3] Department of Genetics and Genomic Sciences, Icahn School of Medicine at Mount Sinai, New York, NY, USA. [4] Summer Undergraduate Research Program, Memorial Sloan-Kettering Cancer Center, New York, NY, USA. [5] Gene Regulation Section, Laboratory of Molecular Biology and Immunology, National Institute on Aging, Baltimore, MD, USA. [6] Department of Physics, Binghamton University, Binghamton, NY, USA. [7] Sorbonne Université, CNRS, Institut de Biologie Paris-Seine (IBPS), Laboratoire Jean Perrin (LJP), F-75005 Paris, France. [8] Center for Communicable Disease Dynamics, Harvard TH Chan School of Public Health, Boston, MA, USA. [9] Howard Hughes Medical Institute and Laboratory of Sensory Neuroscience, The Rockefeller University, New York, NY, USA. ✉email: htm24@cornell.edu; xavierj@mskcc.org

Cooperation between microbes allows microscopic communities to gain strength in numbers and impact their macroscopic environment[1,2]. This can provide huge population-level benefits ranging from antibiotic resistant biofilms[3,4], to fruiting bodies[5] and swarming motility[6]. However, cooperative traits incur a cost if they utilize resources that would otherwise be used for growth[7]. The microenvironments inside bacterial communities are densely packed, dynamic, and highly competitive[8]. How can such cooperative traits persist despite intense competition when they carry a cost to cooperators? Understanding which factors favor selection for cooperative behaviors remains an open question in sociomicrobiology.

Many cooperative behaviors in microorganisms are controlled, at least partially, by cell-cell communication molecules termed quorum signals[9,10]. Cooperative behaviors under quorum sensing control are triggered when a threshold concentration of quorum signals is sensed by the individual. For well-mixed systems in a closed volume, this quorum signal regulation correlates with a threshold population density[2,11]. Regulation of this kind gives a sense of timing to participation in cooperative behaviors whereby bacteria withhold participatory gene expression until the appropriate environmental prerequisites are met. In a system that is spatially-structured or where system volume is unconstrained, communication via quorum signals depends on the mass transfer properties of the relevant molecules and the configuration of the environment[12–14].

Theory states that spatial structure is key in the evolution of cooperation[15]. The gradients of diffusible cues that emerge from the development of the cell populations can lead to localization of the benefits of cooperative behaviors. This can allow the genotype of even a costly cooperative trait to be preserved in a population[16–19]. In practice, additional information such as growth rate can be used to control timing of gene expression so as to minimize any cost to the cooperating individual[20,21] through a regulation mechanism termed metabolic prudence[22]. However, in spatially-structured systems, exactly how gene expression is influenced by the combination of intercellular communication (quorum signals) with intracellular information (growth rate) remains unknown.

*Pseudomonas aeruginosa* is a gram-negative bacterium and destructive opportunistic pathogen capable of several life-threatening virulence behaviors[2]. Moreover, it is a key model organism in the study of microbial social behavior. Their ability to transition between sessile (biofilm) and motile (swarming) lifestyles with the production of rhamnolipids, a biological detergent and virulence factor, is highly conserved[23] and well-studied[6]. This cooperative trait requires the expression of *rhlA*[24,25] and the secretion of massive amounts of rhamnolipids[25,26] that can amount to 20% of the population's dry mass[22]. Swarming allows a colony to grow over an order of magnitude larger in population size compared to mutants (△*rhlA*) unable to produce rhamnolipids[22]. Although the rhamnolipids become publicly available once secreted, the behavior is not susceptible to cheating. Wild type (WT) *P. aeruginosa* do not lose in swarming competition to genotypes that are able to utilize rhamnolipid secretions without contribution[22]. *rhlA* regulation is a key component of this phenotype[22,27,28].

Much of our knowledge of gene regulation in cooperative behaviors has come from studies in liquid culture. However, recent work indicates that gene expression in spatially-structured systems, particularly related to quorum signal communication, may differ from our liquid culture observations[29]. The impact of these claims on cooperative behaviors is impractical to validate with current tools. The methods that exist to interrogate gene expression in spatial structure are either limited to microfluidic installations[12,13], or destroy the sample with data collection, making the observation of dynamical timeseries impossible[30]. New methods that can combine wide-field examination with

time-course observation and high-throughput are necessary. Further, we need new model frameworks to interpret those results. To address these deficiencies, we constructed a fluorescent imager inside an incubator able track cell growth and gene expression directly in cellular communities grown on Petri dishes with high spatial-temporal resolution.

Here we show that spatially segregated colonies of *P. aeruginosa*, previously only reported to show micrometer-scale communication[12,31,32], are capable of responding to communication molecules across centimeter-scale distances. This finding relies on two key aspects of this work: 1—We are able to observe the spatial-temporal development of microbial social behaviors with our custom-built fluorescent imaging assembly by applying slight adaptations to common microbiology techniques. 2—Using this framework, we find that gene expression patterns in *rhlAB* promoter activity in the spatially-structured system differ from those previously characterized in liquid. Further, the behaviors we observe in spatial structure are more predictive of gene expression we observe in motile swarms. Taken together, our results suggest there may be many bacterial phenotypes where gene expression patterns differ between liquid and spatially-structured environments which can already be investigated with minor adjustments to classic microbiology methods.

## Results

**Timeseries imaging tracks gene expression in spatial systems**. Recent studies have shown it possible to identify the members of microbial consortia as well as their gene expression within spatially-structured systems[30,33,34]. However, these methods capture data cross-sectionally and are unable to provide temporal insight into gene expression patterning as it emerges in these cell populations. To bridge this gap, we built a fluorescent imager inside an incubator (Supplementary Fig. 1). Our framework characterizes cellular growth and gene expression in spatially-structured environments with previously unattainable time-resolution and throughput. Fluorescently labeled cells are illuminated using LEDs connected to a custom-built control system (see methods). The images are background corrected and analyzed, tracking colony growth and gene expression information (Supplementary Figs. 2, 3) straight from the spatially-structured system.

In our experiments, we utilized a dual-labeled *P. aeruginosa* PA14 strain harboring PBad-DsRed(EC2)[35] driven by L-arabinose in the plate media, which cannot be metabolized by the cells[36], and P*rhlAB*-GFP[28,37]. When grown in spatial structure, the constitutive expression of DsRed provided a measure of the local density of bacteria (Supplementary Fig. 4). In all our experiments, the dynamical expression of GFP, validated by RT-qPCR (Supplementary Fig. 5) (see methods), reported on the expression of *rhlAB*.

Using these data, we were able to characterize how the surroundings experienced by these microbes influence the dynamics of their cooperative behavior directly in a spatially-structured setting.

**Rhamnolipid production differs in liquid and spatial environments**. Rhamnolipids are necessary for cooperative swarming behavior in *P. aeruginosa* and for other traits related to virulence[26]. Rhamnolipids can be produced in liquid culture[10,20,28,38], thus rhamnolipid production is often studied in detail there. Despite recent work indicating that gene expression related to quorum signaling systems in *P. aeruginosa* may differ in spatial structure[29], no studies assess how downstream genes, such as *rhlAB*, may be affected in spatially-structured colonies. Given the relevance of these diffusible inputs to the *rhlAB* system,

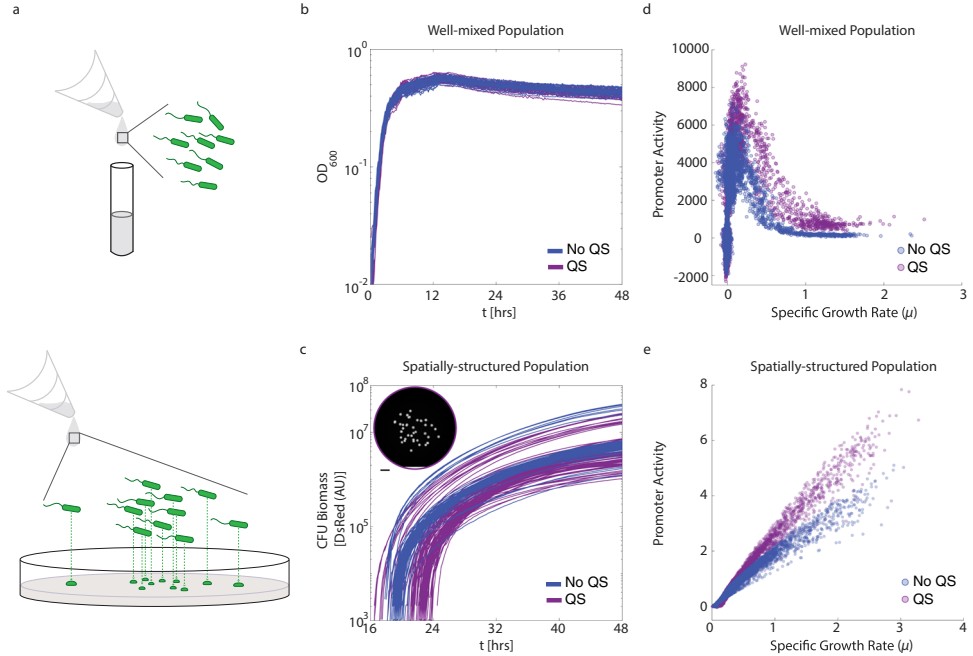

**Fig. 1 Rhamnolipid production differs between liquid culture and surface-attached *P. aeruginosa*. a** Cartoon depictions of liquid and spatially-structured environments used in this study. **b** Optical density timeseries describing *P. aeruginosa* growth in liquid culture. [Blue] Biomass growth without exogenous quorum signals. [Purple] Biomass growth with exogenous quorum signals. **c** DsRed fluorescent timeseries generated from a custom-built imager (Supplementary Fig. 1) and custom software (Supplementary Fig. 3) describing *P. aeruginosa* growth in colony forming units (CFU). [Blue] Biomass growth without exogenous quorum signals [Purple] Biomass growth with exogenous quorum signals added to the plate media. [Inset] Example plate showing colonies at 48 h. Scale bar 1 cm. **d** Promoter activity $\left[\frac{dGFP}{dt} \cdot \frac{1}{OD_{600}}\right]$ of $P_{rhlAB}$ with respect to culture growth rate $\left[\frac{dOD_{600}}{dt} \cdot \frac{1}{OD_{600}}\right]$. [Blue] without exogenous quorum signals [Purple] with exogenous quorum signals. **e** Promoter activity $\left[\frac{dGFP}{dt} \cdot \frac{1}{DsRed}\right]$ of $P_{rhlAB}$ with respect to CFU growth rate $\left[\frac{dDsRed}{dt} \cdot \frac{1}{DsRed}\right]$. [Blue] without exogenous quorum signals [Purple] with exogenous quorum signals provided in the plate media.

we hypothesized that there could be differences between gene expression patterns in liquid and spatial environments.

We compared *P. aeruginosa* biomass growth and gene expression in the liquid and spatial environments (Fig. 1a). Liquid culture data was collected following prior methods[28]. To interrogate the spatial system, we used the protocol from the classic Colony Forming Unit (CFU) assay. Cells were seeded with extreme dilution and we observed the behavior of the resultant colonies (cCFUs) across time and within the random configurations generated.

We observed differences in growth between cells grown in liquid culture (Fig. 1b) and spatial structure (Fig. 1c) with the same media composition. The growth pattern observed in liquid culture recapitulates previously reported data[22,28]. In comparing WT growth (dark blue data in Fig. 1b, c) between environments, we observed that both achieve a period of exponential growth, followed by a period of slowed growth. This sub-exponential growth is prolonged and no period of biomass decay is observed in the spatially-structured environment during our observation window.

Quorum signal perturbation has long been an experimental tool to determine if a phenotype is responsive to social signaling[9,10]. *rhlAB* gene expression in particular is known to be downstream of both the *las* and *rhl* quorum signal systems[39,40]. However, it has previously been shown that liquid culture perturbation with additional C4-HSL and 3-oxo-C12-HSL, the *rhl* and *las* quorum signal system auto-inducers respectively, do not illicit significant change in growth or $P_{rhlAB}$ dynamics in this strain of *P. aeruginosa*[22]. We replicated this liquid culture result (Fig. 1b, purple data). In the spatially-structured system, we performed this perturbation by including both quorum signal molecules in the plate media in the same concentration by volume as previously

published[22]. This analysis was done using biological replicates with <70 colonies (Fig. 1c [Inset]). In comparing between colonies grown with or without quorum signals in the plate media, we observed that colonies perturbed by quorum signals may achieve a smaller final size after 48 h of growth (Supplementary Fig. 6a). We did not observe a difference between the specific growth rate of the colonies during the time interval when they came above detection (Supplementary Fig. 6b). However, we did observe that colonies given quorum signal perturbation show later colony detection (Supplementary Fig. 6c).

We analyzed the promoter activity[28,41] of *P. aeruginosa* grown both in liquid culture (Fig. 1d) and spatial structure (Fig. 1e). In liquid culture, we found *rhlAB* promoter activity to be low during periods of high specific growth rate as seen previously[22,28]. Promoter activity increased as the specific growth rate decreased and below a threshold growth rate promoter activity dropped as expected during prolonged stationary phase[28]. Unexpectedly, in spatial structure, we observed a strong positive correlation between specific growth rate and promoter activity ($R^2 = 0.96$) (Fig. 1e).

Previous work done in liquid culture captured no significant change in rhamnolipid production in WT bacteria grown with quorum signals added to the media[22] and our data agree (Fig. 1d, purple data). Conversely, we found that in spatial structure, WT colonies expressed even higher levels of *rhlAB* during periods of high growth rate when quorum signals were added to the same plate media recipe (two-sided rank sum test *p*-value < 1e−4, see methods). This presents in our data as a steeper positive slope in the association between specific growth rate and promoter activity ($R^2 = 0.98$). We conclude that not only are there phenotypic differences between *rhlAB* gene expression in the liquid and spatial systems, but that there

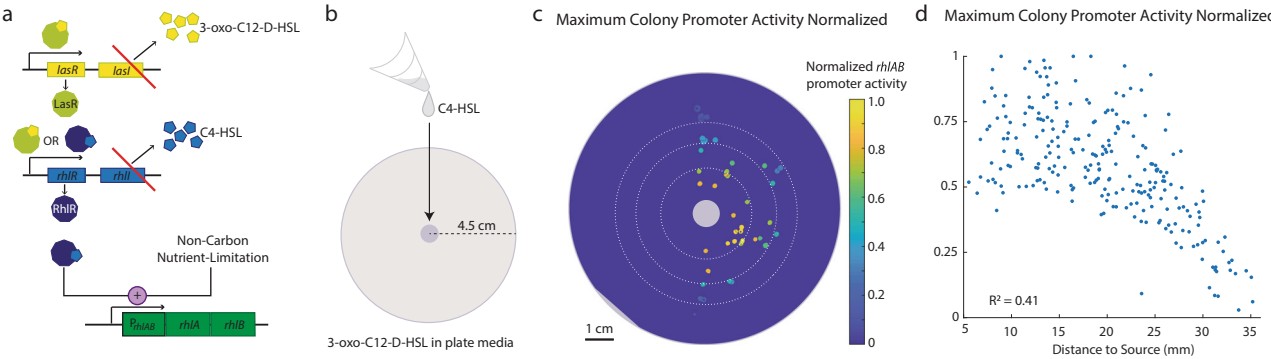

**Fig. 2 *P. aeruginosa* responds to diffusive quorum signals in a distance-dependent manner. a** The native molecular circuit determining *rhlA* expression. Red lines describe clean deletions present in a signal mute quorum signal mutant. **b** Experimental design to investigate the length scale of quorum signal response. **c** Colonies viewed at their 24 h size colored by the maximum promoter activity achieved across the colony's timeseries. Data has been normalized. **d** Maximum promoter activity achieved by each colony plotted with respect to the distance between the center of the colony and the center of quorum signal source. Data from three independent biological replicates is shown, with 40, 69 and 139 colonies respectively. Data are normalized to the highest promoter activity observed for the dataset to correct for batch effects.

is a phenotypic difference under quorum signal perturbation that is specific to the spatially-structured system.

**Cellular response to diffusive quorum signals is distance-dependent.** Next, we sought to understand whether the differences we observed in liquid culture and spatially-structured gene expression could be driven by diffusible quorum signals. Previous work in liquid culture revealed that *rhlAB* expression can integrate nutrient and quorum signal information from at least three diffusible small molecules: a growth-limiting nutrient[22,28] and the hierarchical quorum sensing pathway involving the autoinducer molecules 3-oxo-C12-HSL and C4-HSL[40,42–46]. When bound to their cognate receptors, LasR and RhlR respectively, these complexes may act as transcription factors, instigating systemic gene expression change[47,48] (Fig. 2a). All three diffusive inputs are of similar molecular size and thus may act on similar length and time-scales. The ratio of the diffusion coefficients and decay rates for these molecules in bacterial growth media[38] indicate that the quorum signals could reach biomass that is multiple millimeters away, though whether their physiologic concentrations could influence biomass at that distance was unknown.

*P. aeruginosa* has been shown capable of micrometer length-scale communication in constrained microfluidic experiments[31]. However, it is difficult to extrapolate these results to the full spatial-temporal system. With the knowledge that *P. aeruginosa* cCFUs respond to systemic quorum signal perturbation, we asked: over what macro- spatial-temporal scales are these cells capable of responding to quorum signal perturbation? To address this, we utilized a signal-mute mutant, PA14 Δ*lasI*Δ*rhlI*, that cannot produce the 3-oxo-C12-HSL and C4-HSL molecules, but is able to respond when these signals are exogenously provided (Fig. 2a). In these experiments, this strain was double-labeled in the same way as the WT PA14. We focused on response to C4-HSL. We added the upstream quorum signal (3-oxo-C12-HSL) directly to the plate media and loaded 4 μL of 5 μM C4-HSL on a filter paper in the center of the plate (Fig. 2b).

We tracked the growth and *rhlAB* expression in colonies seeded around the filter paper (Fig. 2c). In this experimental configuration, observing P$_{rhlAB}$ activity in a colony indicated that it had encountered both quorum signals at concentrations high enough to trigger a *rhlAB* response. We found that the response in the signal-mute mutants varied with the distance between the colony and the center of the filter paper (Fig. 2c, d).

We found that maximum colony promoter activity was inversely proportional to the colony's distance to the filter paper (Fig. 2d) ($R^2 = 0.41$). The highest maximal promoter activities we observed occurred in colonies <2.5 cm from the quorum signal source. In colonies between 2.5 and 4 cm away, the maximal promoter activity scaled with the distance to the quorum signal source more strongly ($R^2 = 0.63$). We investigated the presence of rotational biases in our data by comparing the distributions of maximal promoter activity within 45° increments with the two-sided rank sum test (Supplementary Fig. 7). The strongest bias our investigation revealed was between 90 and 135°. However, in our dataset, this region had fewer samples (Supplementary Fig. 7b) and all colonies were within 2.75 cm from the quorum signal source (Supplementary Fig. 7c), a region of high variability across all our data.

These experiments carried out with the signal-mute mutant confirm that *P. aeruginosa* can respond to diffusible quorum-signal perturbation on a centimeter length-scale. They also illustrate that *P. aeruginosa* is capable of a concentration-dependent dose-response to diffusible quorum signals. As our experimental protocol uses physiologically relevant quorum signal concentrations and time-scales[22], these data indicated that these cells may be capable of configuration-dependent behavior.

**Inference of the cellular spatial environment by model selection.** Given our results suggesting that spatially distinct multicellular aggregates may be capable of communication over macro-scale distances, we next looked to test whether similar centimeter length-scale interactions could be observed in the WT. The diffusion coefficients for the quorum signals we have investigated here, C4-HSL (MW 171.9 g/mol) and 3-oxo-C-12-HSL (297.37 g/mol), are on the order of ~$7 \times 10^{-6}$ cm$^2$/sec, slightly slower for the larger 3-oxo-C-12-HSL[49]. This means that over the course of 24–48 h, these signals are capable of traveling 1–2 cm away from their source. Based on this, we predicted that *P. aeruginosa* colonies may be able to detect and respond to each other within small macro-scale distances within the viewing timeframe of our experiments. However, the integration of these signals in liquid is known to be non-trivial[20,22,28]. This infrastructure poses a system that may be highly sensitive to fluctuations in the diffusive environment. Therefore, a data-driven and unsupervised approach was required to provide unbiased insight into the spatial-temporal scale over which spatially segregated bacterial communities may influence one another.

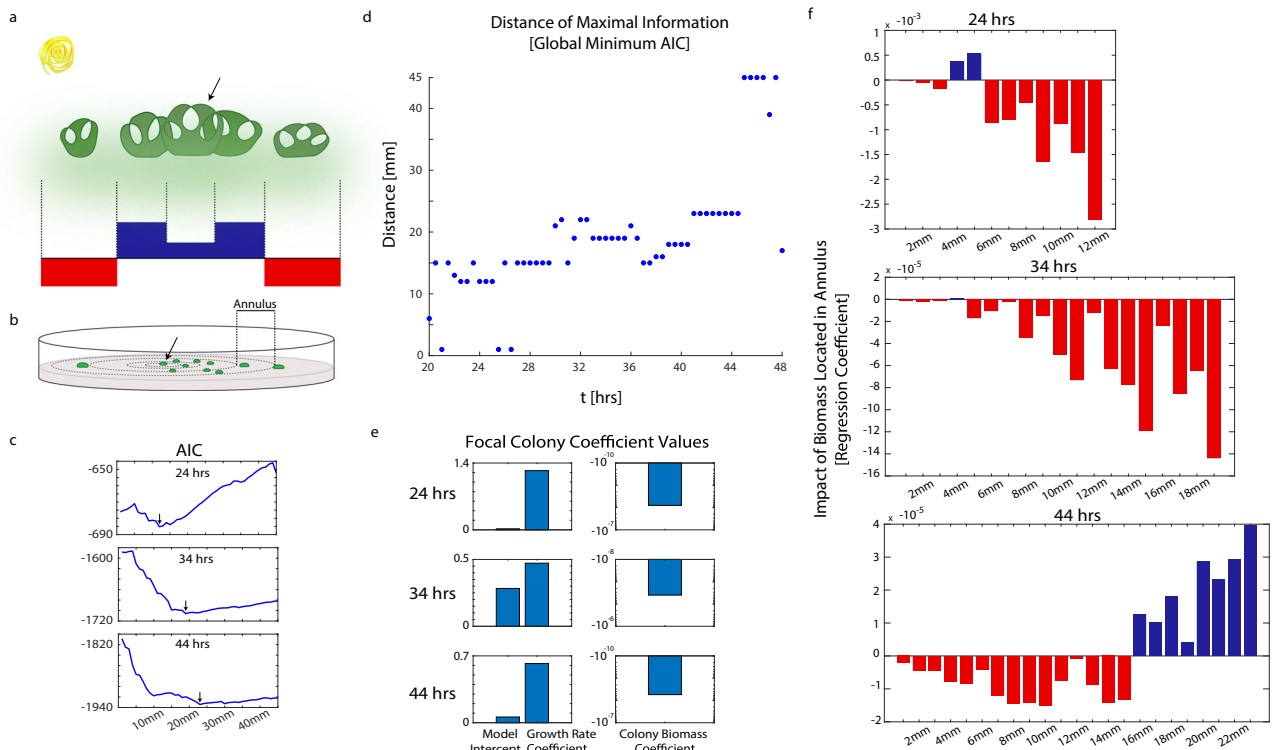

**Fig. 3 The spatial-temporal environment of a colony as inferred by statistical model selection. a** Cartoon describing spatial kernels as identified in arid ecological systems. The growth of a focal individual (black arrow) can be influenced by its surrounding environment. **b** Cartoon depicting a spatial kernel framework in a microbial CFU experiment. Black arrow indicates focal colony. **c** Model selection results, quantified by the Akaike Information Criterion (AIC). Each point on the curve represents the AIC of a model created with data at the timepoint specified. The features in the model include colony-centric features (see **e**) and all annuli proceeding outward from the focal colony up to and including the distance indicated along the *x*-axis. The minimum AIC is marked (black arrow). **d** Distance corresponding with the best regression model (global minimum AIC) at each timepoint. **e** Intercept and focal colony coefficient values fitted independently at three timepoints. **f** Spatial-temporal kernel models. Each ecological kernel is fitted independently at the designated timepoint.

To test these predictions, we performed CFU experiments with between 60 and 150 WT colonies seeded throughout the plate. Here, the variability in colony location inherent to extreme dilution seeding provided an experimental advantage. These plates explore a variety of colony configurations. Each colony had a unique location with respect to every other colony on the plate. Each biological replicate had colonies spread over a similar total plate area. As a result, this approach generated a large amount of variation in growth pattern and gene expression. This self-generated variation allowed us to leverage unbiased and data-driven methodology to uncover, 1—whether colony-colony interactions could explain the variation we observed in colony growth and gene expression and if so, 2—how did these interactions develop with time and over what spatial scale? We define colony-colony interactions here by statistical association, examining whether focal colony promoter activity can be explained by the growth patterns of non-focal colonies.

To answer these questions, we applied a spatial kernel approach—a method long used in biophysical systems to model the patterns generated by the interactions between spatially segregated or dispersing individuals[50–53]. As an example (Fig. 3a), arid landscapes often show a patchy pattern of vegetation due to the presence and removal of limiting resources such as water and soil nutrients[53]. A single shrub in an arid environment can preserve some water and nutrients in the soil surrounding their roots. Other shrubs in close proximity impact the focal individual positively. By making the local root system dense, the subsequent preservation of nutrients facilitates the survival of all nearby plants. However, shrubs farther away have a competitive

(negative) impact, drawing nutrients away from the localized hub[54,55]. This methodology can capture behaviors occurring simultaneously across multiple length-scales while allowing the flexibility to encapsulate a wide range of shapes[51] besides the Gaussian shape typical of diffusional processes. We apply this same idea to describe the positive and negative effects that the spatial configuration of biomass may have on the cooperative behavior of a focal colony. However, where previous work has used the data to fit the parameter values for spatial kernels of a specified shape[50,51,56], we use a data-driven approach to fit the shape of the spatial kernel itself. This is the value of the heterogeneity generated by our application of the CFU assay protocol. The variety in spatial configuration allows us to sample a wide range of colony arrangements, giving us the statistical opportunity to infer the spatial-temporal length-scales at play directly from the data.

The spatial kernel can be modeled as a collection of concentric annuli of fixed radius emanating from each colony (Fig. 3b). The promoter activity of a focal colony (Fig. 3b, black arrow) was investigated with respect to the surrounding colony configuration. We discretized the kernel with a 1 mm distance between the outer and inner radii of each annulus. We fit the following linear model to the colony promoter activity:

$$P_{C_F}(t) \sim \mu_{C_F}(t) + B_{C_F}(t) + \sum_{i=1}^{A} \sum_{j=1}^{n_i} B_{C_{i,j}}(t) \qquad (1)$$

where $P_{C_F}$ is the promoter activity of a focal colony, $\mu_{C_F}$ is the specific growth rate of the focal colony and $B_{C_F}$ is the amount of

biomass in the focal colony at time $t$. $B_{C_{i,j}}$ is the amount of biomass in colony $j$ in annulus $i$ where there are $A$ total annuli and $n_i$ colonies in the annulus of interest. In this formulation, focal colony promoter activity may scale with the colony's growth rate as well as the number of cells present in the focal colony at time $t$. The total biomass in each annulus is used as a series of features in the model to explain the variation in promoter activity between colonies of similar size and growth rate. All annulus features were normalized for annulus area before fitting to ensure that all annuli contributed equally to the fit. We assumed that all cells that founded a CFU landed on the plate at the same time. Diffusion processes that impacted a colony were then assumed to occur across the same time-scale for all colonies simultaneously. It stands to reason that a focal colony may experience a time-delay in the impact of distant colonies; we found our implementation to be a decent approximation.

We independently fit models for data taken every 30 min between 20 and 48 h. At each timepoint, a series of models were fitted where each new model included one annulus farther from the focal colony than the previous model. We compared models using the Aikake Information Criterion (AIC) to assess the trade-off between model simplicity (fewer features) and the quality of fit. In accordance with standard practice, we chose the best model as the one with the lowest AIC. We interpret the distance of the largest annulus included in that optimal model as the longest length-scale of colony-colony interaction at that timepoint (Fig. 3c).

Our AIC-selected spatial kernel approach fit our data well (Supplementary Fig. 8) and revealed the surprisingly clear result that the colony-colony interaction length-scale lengthens with time ($R^2 = 0.54$) (Fig. 3d). Further, focal colony feature coefficients showed internal consistency across our independently fit models. (Fig. 3e). Model selection identified interactions between colonies up to 1–1.5 cm apart early in the timeseries (22–26 h) and up to 1.8–2.3 cm apart later in the timeseries (40–44 h). This data-driven approach leads us to conclude that WT PA14 are capable of centimeter length-scale colony-colony interactions within a 48 h window.

Finally, we reviewed the spatial kernels predicted by our model to see how biomass localized in each annulus was predicted to influence focal colony promoter activity at various timepoints. Earlier in the timeseries, at 24 h, we observed that biomass more than 0.5 cm away from the focal colony negatively impacted promoter activity. However, this relationship shifted with time. By 44 h, all colonies within 1.5 cm of the focal colony had a negative impact, while colonies more than 1.5 cm away may have positively impacted focal colony promoter activity (Fig. 3f). We do not claim that these interactions are due only to the diffusion of C4-HSL and 3-oxo-C12-HSL, though these results do match the length-scales of interaction predicted by our investigation of quorum signal gradients (Fig. 2). All together, these results characterize a general length-scale of gene expression association between colonies on the order of 1–2 cm that lengthens and changes shape with time.

**Swarm tendrils achieve exponential growth despite constant velocity.** Uncovering associations between gene expression patterns in spatially-distinct biomass aggregates led us to ask whether these findings could extend to motile *P. aeruginosa* swarms. This swarming behavior has long been of general interest due to its example as a cooperative behavior that is not invadable by non-cooperating strains in competitive assays[22]. Given our success in interrogating gene expression directly in spatially-structured systems, we looked next to extend our investigations to the motile swarming system.

Specifically, we wanted to know whether the WT motile swarms would show growth and *rhlAB* promoter activity patterns more similar to classic well-mixed liquid culture or the new dynamics found in the immotile spatial system. To do this, we fluorescently imaged swarms (Supplementary Fig. 9) and isolated cross-sectional biomass (DsRed) and P$_{rhlAB}$ (GFP) measurements along the length of three tendrils in each of four independent swarms. We first investigated growth in swarming tendrils (Fig. 4a, b[top]). In a striking departure from both our liquid and cCFU results, we found that despite attaining an average constant velocity of 3.56 mm/h (±0.65 mm/h), these tendrils were capable of achieving and sustaining periods of exponential growth (Fig. 4a, dashed line—linear growth trajectory). This phenomenon may be related to cell motility as seeding cells in a tendril configuration on motility-preventing agar showed growth dynamics similar to cCFUs (Supplementary Fig. 10).

**Swarming tendril P$_{rhlAB}$ activity matches cCFUs.** We next looked to compare the relationship between promoter activity and growth rate within a swarming tendril. To do this, we calculated these metrics along our tendril cross-sections and examined the data for spatial localization (Fig. 4b). We separated our cross-sections into three segments: the swarm center, the swarm edge, and the mid-tendril region between them (Fig. 4c inset). The edge of the tendril was a region that showed biomass localization, located near the tip of each tendril (Fig. 4b, Supplementary Figs. 11b and 12). It was typically between 2.75 and 4.5 mm in length. This analysis again revealed a positive correlation between growth rate and promoter activity $R^2 = 0.79$ in the tendril tip, the front-most 0.86 mm of the tendril edge (Fig. 4c). This finding continues to be highly counter-intuitive given previously published work[22,28] as well as our own liquid culture data (Fig. 1b, d). However, these trends fit with our new spatially-driven expectations (Fig. 1c, e). We performed our analysis conservatively, only examining a pixel after biomass had been present in that location for three or more timepoints (15 min) to prevent artifacts. This approach did not account for biomass flux into or out of any pixel. We assumed that the mid-tendril region is seeded by cells left behind as the edge proceeds away from the swarm center. As there will be non-negligible flux out of the tendril edge, the growth rate calculated for the edge of a tendril may be an underestimate. By contrast, the mid-tendril and the swarm center exhibited a much narrower range of growth rate and corresponding promoter activity (Fig. 4d, e).

**Quorum signal perturbation reveals swarm biomass redistribution.** Finally, we wanted to know how perturbation with quorum signals impacted swarming behavior. In light of our previous results, we hypothesized that in swarms provided with exogenous quorum signals we would see higher promoter activity related to rhamnolipid production, resulting in faster spreading and earlier tendril formation in *P. aeruginosa* swarms.

We extracted data from three tendrils in each of three independent swarms grown with quorum signals in the plate media and examined the data for quorum signal-induced phenotypic differences. We found that, similarly to the non-perturbed swarms, these swarms were able to achieve a sustained period of exponential growth along the length of each tendril and reach the same total biomass at the end of our 24 h timeseries (Supplementary Fig. 11).

Surprisingly, we found that localization of biomass within these tendrils differed significantly from the original WT swarming tendrils (Supplementary Figs. 11–12). We found that under quorum signal perturbation, biomass increasingly localized to the swarm center and the tendril edge as the tendrils lengthened

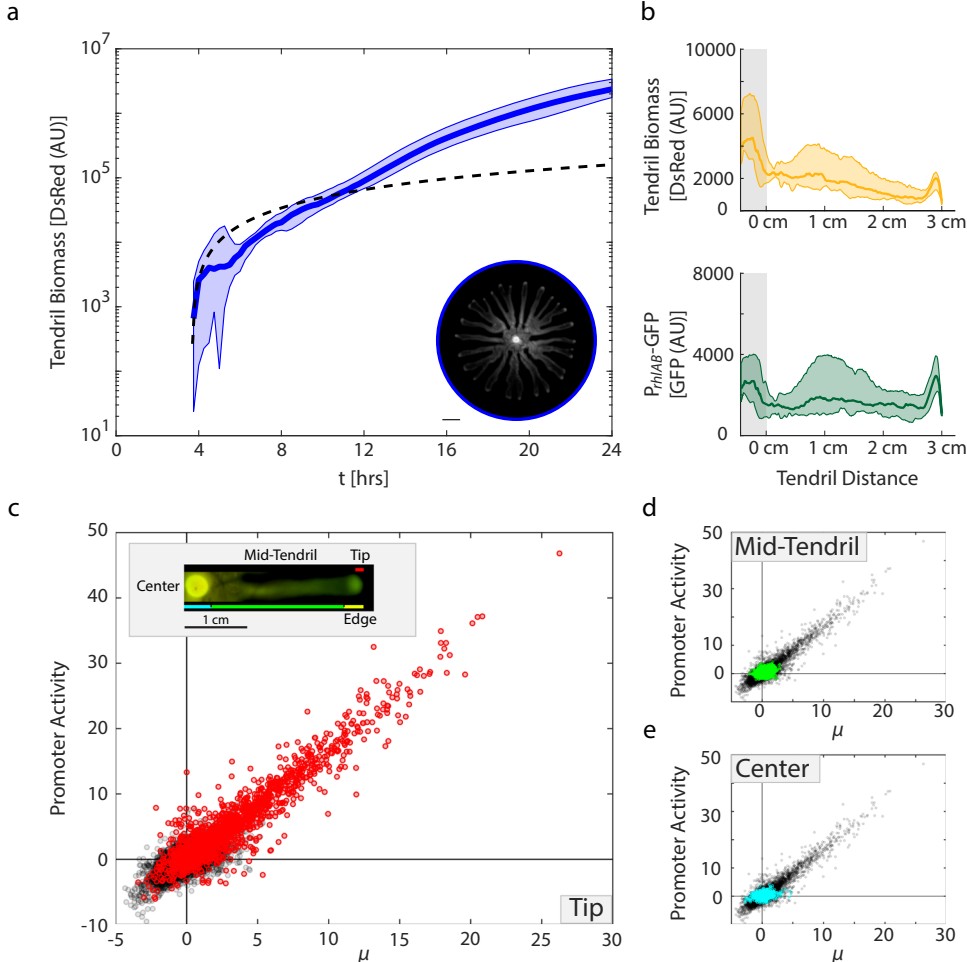

**Fig. 4 Swarms show behavior similar to immotile colonies grown in hard agar. a** Biomass in WT swarming tendrils over time (12 tendrils). Line indicates median data. Full range of data shaded. Data has been smoothed with a moving window of 5 for visualization. [Inset] Image of swarm. Scale bar 1 cm. **b** Biomass and GFP distribution in tendrils 3 cm in length. Gray bar indicates swarm center. Shaded regions indicate the full range of the data. Middle line indicates median data. **c** [Inset] Coloration legend. Image of a WT swarm tendril with relevant region delineations indicated. [Main Panel] Promoter activity in each pixel with respect to growth rate during the corresponding time interval. Red coloration indicates that the pixel was in the tendril tip, the 0.86 mm closest to the front edge of the tendril. **d**, **e** Same data as in (**c**), pixels in cyan originated in the swarm center, green pixels originated between the swarm center and tendril edge. **c**–**e** Show data for a representative tendril.

(Fig. 5a). To our knowledge, a socially-driven spatial segregation phenotype such as this has not been previously identified in *P. aeruginosa*, with the closest comparisons being the social regulation involved in facilitating fruiting body formation in *Myxococcus xanthus*[57] or *Bacillus subtilis*[58].

We found that, indeed, swarms provided with quorum signals form tendrils sooner than non-perturbed swarms, *p*-value < 1e−8 by Kolmogorov–Smirnov test (Fig. 5b), and moved with a faster average velocity (4.58 mm/h ± 1.00 mm/h) than swarms not given quorum signals (3.56 mm/h ± 0.65 mm/h). Cells in the tendril tips of these swarms achieved higher growth rates, *p*-value < 1e−10 (Fig. 5c) and higher promoter activities (Supplementary Fig. 13), scaling linearly with growth rate, $R^2 = 0.89$. We did not detect a change in the slope of the relationship between promoter activity and growth rate in *P. aeruginosa* swarms perturbed with quorum signals.

**Quorum signal perturbation does not facilitate invasion by defectors.** Lastly, we investigated swarming competition in the presence of quorum signals. Rhamnolipid production has long been posited as a possible competitive weak point in *P. aeruginosa* cooperative swarming as it represents a large resource investment. Once secreted, the rhamnolipids can be utilized by other cells in

the vicinity[22]. However, competitions conducted between the WT and a rhamnolipid defector (Δ*rhlA*) have never shown the WT to definitively lose. In light of our quorum signal perturbation data in swarms (Supplementary Fig. 13), it was unclear whether the increased P*rhlAB* activity could make these cells susceptible to invasion by this defector strain in a competitive setting.

Swarming competition experiments were performed as previously reported[22], now with plate media containing quorum signals. The exact initial mix of WT to defector was calculated to account for variability due to mixing and dilution $\left(\frac{WT_i}{Total\ Cells_i}\right)$. After the competition, the final ratio $\left(\frac{WT_f}{Total\ Cells_f}\right)$ was calculated. We found that, in our hands, the WT continued to resist invasion by the defector strain despite quorum signal perturbation (Fig. 5d).

## Discussion

Here we characterized the expression of a gene for a microbial social behavior in a spatially-structured environment. Our fluorescent imaging approach allowed the tracking of biomass growth and gene expression in space and time. With it, we

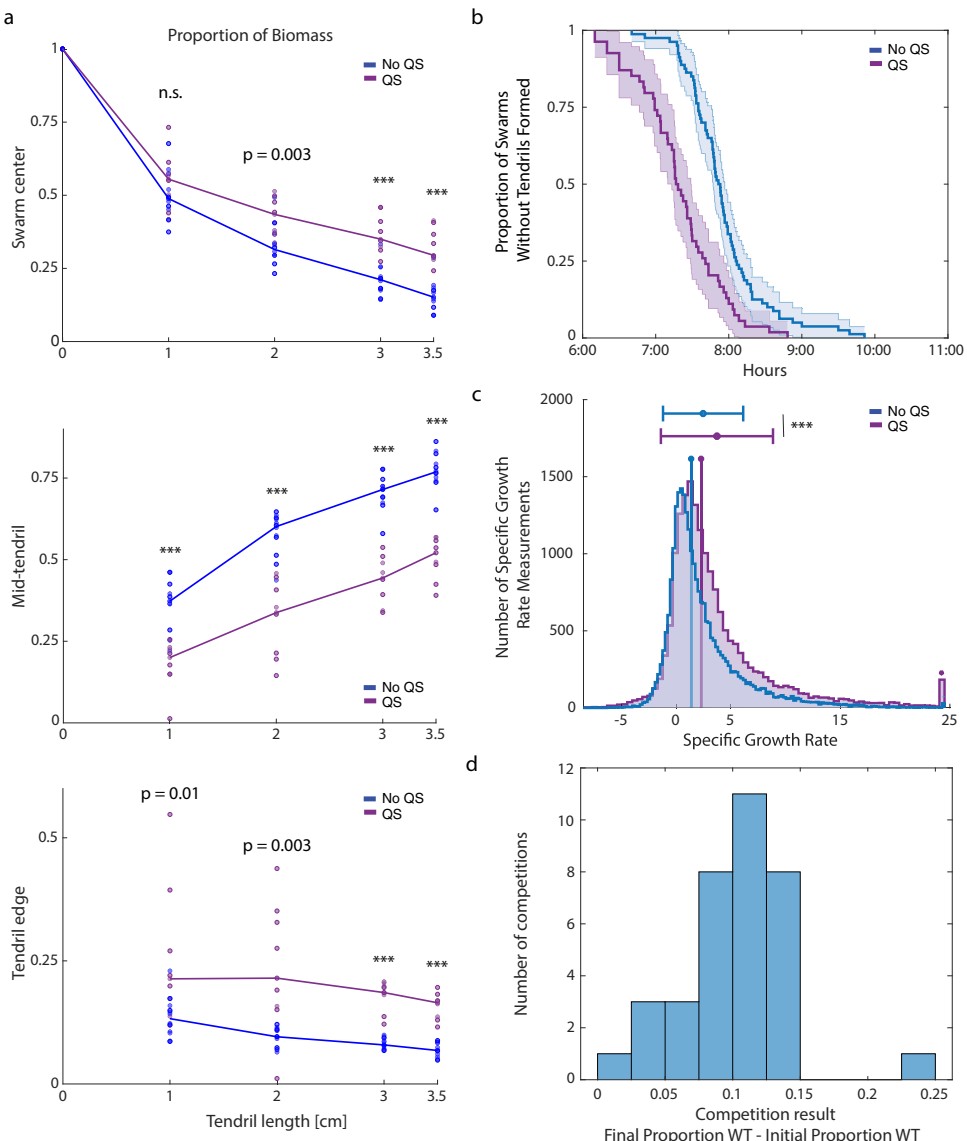

**Fig. 5 Quorum signal perturbation reveals accelerated tendril formation, biomass redistribution, and no competitive disadvantage. a** Proportion of all tendril biomass along tendril length localized to the indicated regions of a swarm tendril defined graphically in Fig. 4c: Swarm center (top), mid-tendril region (middle), and tendril edge (bottom). The tendril edge is defined as the outermost 4.5 mm of a swarming tendril relative to the swarm center. All statistical comparisons performed with the two-sided rank-sum test. *** indicates a *p*-value < 0.001. Results where the null hypothesis was not rejected are indicated by n.s. **b** Swarming start times with or without autoinducer in the plate media. **c** Specific growth rate in the tendril tip (outermost 0.86 mm of swarming tendril) in swarms without (left) and with (right) quorum signals provided. The two distributions were compared using the two-sided rank-sum test, *p*-value < 2.22e−16. Stem plot indicates distribution median. Mean and standard deviation indicated above histogram. Histogram bar with (*) contains all points with specific growth rate greater than maximum *x*-axis value. **d** Competition results for WT PA14 against the Δ*rhlA* strain in 1:1 ratio with quorum signals in the plate media (see methods). Data includes three biological replicates with several technical replicates in each. See Supplementary Table 1.

uncovered striking gene expression patterns not previously observed in liquid culture. We confirmed that in liquid culture, P*rhlAB* activity was low at high cellular growth rate and did not respond to quorum signal perturbation (Fig. 1d). However, in spatially-structured cCFUs, we found that promoter activity scaled positively with growth rate and cCFUs responded to quorum signal perturbation (Fig. 1e). Using a quorum signal mute mutant, we characterized a continuous, distance-dependent response to a diffusible quorum signal (Fig. 2).

We then applied a spatial kernel framework to derive a data-driven statistical model to characterize the spatial-temporal length scales of bacterial gene expression in the spatially-structured environment. We characterized interactions between colonies 1–2 cm apart, the largest distance of inter-aggregate

communication reported to-date. Our model predicted that the interactions between neighboring colonies are largely inhibitory (Fig. 3f), which may represent nutrient competition. However, at late timepoints, our model predicted a positive impact of biomass between 1.6 and 2.3 cm away from the focal colony. While the cause is unknown, this may be a response to the arrival of quorum signals from a neighboring colony stimulating additional promoter activity.

Investigating swarms in our fluorescent timeseries framework revealed that swarming facilitated a surprising exponential growth trajectory despite a linear tendril velocity (Fig. 4a). These data indicate that this system may be an example of navigated range expansion, following recent observations of growth dynamics in motile *E. coli* strains[59].

We tested our findings in the competitive environment, curious whether the higher promoter activity achievable in the quorum signal-perturbed swarms could create an opportunity for invasion by a defecting strain. It was not, leaving us to conclude that even in a quorum signal high background, the WT is able to control its investment into rhamnolipid production to resist invasion by defectors. This result is in line with the competitive roots of metabolic prudence[22], despite the apparent differences in implementation (Figs. 1d, e, 4c). If there is a regime of specific growth rate that shows low promoter activity as in liquid culture, it would need to occur at growth rates higher than the dynamic range we observe and thus would likely occur at densities too low for us to observe. Considering we also do not observe this phenomenon in the swarming tendrils (Fig. 4c), even if the dynamics can occur, they may not occur in a regime that is relevant for swarming competition.

The coupling we observe between promoter activity and growth rate may provide a regulation mechanism itself to prevent invasion by rhamnolipid defectors that is inherent to the spatial environment. Cells at the tip of a swarming tendril are exposed to new nutrient sources and produce copious rhamnolipids corresponding with the cellular specific growth rate (Fig. 4c). This facilitates the continued outward progression of the tendril. In a competitive environment, if the cells are evenly mixed with a defector strain, then fewer cells at the tip of a tendril are producing rhamnolipids and tendril speed could drop as a result. The depressed rate of expansion would constrain the availability of fresh nutrients to the cells in the tendril tip, leading to lower specific growth-rates and with a corresponding drop in rhamnolipid production (Figs. 1e, 4c) and the swarm tendrils will not reach the edge of the plate within 24 h[27]. This mechanism would fit with the data from Xavier et al. 2011[22].

In our quorum signal perturbed swarming competitions, it may be that despite even mixing at the start of the competition, the biomass redistribution observed with quorum signal perturbation (Fig. 5a) is enough to spatially segregate the two phenotypes, though such segregation has not been reported. Regardless, these data suggest that there are more alterations to gene expression, or perhaps cellular metabolism, to be uncovered in this system in the presence of quorum signals in a spatially-structured environment.

As we expand gene expression studies into spatially-structured systems using newly available cross-sectional tools[30,33,34], there is no replacement for approaches such as ours to link comparisons between biological replicates to observation of a system directly with high time-resolution. We present this work as a blueprint for a data-driven approach to capture and characterize behaviors similar to these in new environments where complex interactions between diffusible inputs is expected. As many social behaviors take in diffusive inputs, these results may be generalizable to a wide range of social or cooperative phenotypes with spatially-linked gene regulation that can already be assayed with classic microbiology techniques.

## Methods

**Microbiological assays**. Plates were made with the recipe from Supplementary Table 2[6] with the addition of L-arabinose at 40% weight/volume for a concentration of 1.5%. Water was subtracted to compensate. Every plate had 20 mL of agar media and was inspected visually to confirm a flat surface. Swarms were prepared by spotting 2 µL of a triple-washed cell culture onto the center of a petri dish filled with swarming agar[6]. Liquid culture assays were performed using the same media recipe as the agar plates, replacing agar with water[22] and data were acquired on a benchtop TECAN M1000 plate reader. Unless noted otherwise, swarms and colonies provided with exogenous quorum signals were given the concentration of quorum signals determined to be present after 24 h of swarming[22]. See Immotile Colony Analysis for further details on cCFU assays.

**Timeseries analysis**. All timeseries were imaged with our prototype imaging device (Supplementary Fig. 1). Fluorescent LEDs were used to light the sample. Data were collected by an Atik VS14 Fluorescent Camera through the Thorlabs

filter wheel FW102C. Timeseries were collected through a custom-built control system using the Arduino Uno R3. Immotile colony timeseries were imaged every 10 min. Swarms were imaged every 5 min. Images taken were subject to uneven lighting due to the placement of the fluorescent LEDs (Supplementary Figs. 1, 2). To correct for this, multiple plates were imaged after timeseries collection on a GE Typhoon Trio flatbed fluorescent scanner. We called this scanner data our 'ground truth'. A correction was built from these images that allowed us to take each image generated in our device and convert it, simulating the evenly lit environment on the scanner. This correction was built manually by extracting features from the images. The data were validated on rotational datasets (Supplementary Fig. 2). The final background correction is shown below. Parameters vary depending on the exact configuration of our device though the terms remain consistent. The correction was updated as the instrument received upgrades and to control for variation in the L-arabinose batch used.

$$R_{Scanner} = a + bx + cy + dxy + eR_{Colony}(1 + fx + gy) + h(Distance\ To\ Plate\ Center)(1 + iR_{Colony}) \quad (2)$$

The data were analyzed using custom software in MATLAB 2018a. Liquid culture data was analyzed using custom software for timeseries analyses with experimental replicates[28]. Analysis of spatial-temporal data used to investigate colony-centric growth, promoter activity and all other spatial-temporal features, was developed for this study (Supplementary Figs. 2, 3). Background corrected pixel data was smoothed once with a moving window of 5 along the time axis before pixel data was extracted and grouped into colony or swarm tendril data. Colony DsRed data was smoothed once with a moving window of 5 before analysis and before colony exponential growth rate and promoter activity were calculated.

The GFP fluorophore in the construct used here has been shown to provide differential gene expression information in liquid culture and spatial structure, making it the best model to capture *rhlAB* activity in spatial structure with our new level of quantitative detail[22,28,37]. We repeated this validation and extended it to spatial structure in this work (Supplementary Fig. 5) (see qPCR validation). However, this fluorophore created a halo effect around cell aggregates producing GFP in our spatially-structured system (data not shown). This effect made it difficult to determine the borders of cell aggregates using GFP alone. We adjusted for this by using the DsRed data, which does not generate a halo effect, as a marker to indicate the localization of biomass. In all cases, colony (cCFU or swarming) biomass localization was determined by the DsRed fluorescence only. GFP data was analyzed only where the DsRed fluorescence indicated that cell biomass was present.

Whenever data are shown with a median and shaded region, this depicts the median data with the full range of the data shaded (Figs. 4ab, 5a, Supplementary Figs. 10b, 11, 12). In certain cases, the minimum and maximum data of the full range (Fig. 4ab, Supplementary Figs. 10b, 11, 12) was smoothed once with a moving window of 5 (timepoints or pixels as was applicable) for visualization.

**RT-qPCR validation**. To validate our GFP reporter fusion in liquid culture, extending previous validation analyses[60], we utilized an established growth curve dilution framework[37] that allowed us to extract cells at various stages of growth at a single timepoint. In this way we could assay a wide range of *rhlAB* and GFP mRNA levels for analysis. Cells were grown in a TECAN M1000 plate reader at 37 °C with shaking. For all samples, a matched sample at the same dilution in a 3 mL volume was grown in an incubator at 37 °C with shaking. At 24 h, all samples were harvested. All technical replicates grown in the plate reader were combined into a single sample for RNA extraction. All 3 mL samples were used in a paired extraction. RNA was extracted using the Qiagen RNeasy Kit. RNA was converted to cDNA using the ThermoFisher SuperScript™ IV VILO™ with EZDNase Kit. qPCR was performed using Kapa Biosystems Kapa Sybr Fast—Sybr Green Kit. For primers used see Supplementary Table 3[60].

To investigate differential gene expression in our P*rhlAB*-GFP construct in spatial structure, we utilized the PA14 ΔlasI ΔrhlI P*rhlAB*-GFP and P_BAD-DsRed strain. Cells were grown on agar plates with 1 µM 3-oxo-C12-D-HSL in the plate media. Each plate contained a different concentration of C4-HSL in a logarithmic titration from 0 to 0.5 µM. This experimental infrastructure allowed us to tightly control the quorum signal environment these cells encountered and thus generate dose-response data across a wide range of gene expression levels. DsRed and GFP levels were measured using a GE Typhoon Trio flatbed fluorescent scanner after 24 h and RNA extraction followed.

qPCR analysis was performed by the Delta-Delta CT method compared against housekeeping gene *proC* as in[60]. Primers for *rhlAB* as well as GFP were used in our protocol. To validate our reporter, we first compared *rhlAB* and GFP relative gene expression levels as determined by qPCR. We found that they matched in both liquid and spatial structure data (Supplementary Fig. 5a, d) with $R^2 = 0.96$ and $R^2 = 0.93$ respectively. We then compared the relative gene expression of GFP and *rhlAB* to our measurements of per cell gene expression in each respective experimental system. In liquid culture, we compared relative gene expression of GFP and *rhlAB* to GFP/OD and in spatial structure we compared to GFP/DsRed data. We found that in both systems our relative gene expression correlated well with our real-time reporter measurements (Supplementary Fig. 5b, c, e, and f) with $R^2$ values of 0.78, 0.80, 0.94, and 0.93 respectively. All $R^2$ values were calculated by comparing the cycle threshold

for the gene of interest relative to our housekeeping gene (ΔCT) with the corresponding real-time reporter data. We visualize our qPCR data using the fold change gene expression on a log scale for interpretability.

**Analysis of immotile colonies**. Cells were grown overnight in Casamino acid media, passaged into fresh Casamino acid media and taken for use from exponential phase. The cells were then triple washed and diluted in PBS to between ~0 and 5 CFU per 15 μL. Colonies were plated with motility-preventing agar concentrations as in the classic CFU assay (Fig. 1a, c[inset]). Image timeseries began after the droplets on the plates dried, before the cells were detectable by fluorescent imaging.

The cells were fluorescently labeled for both biomass generation and rhamnolipid investment. Biomass was tracked using DsRed(DC2)[35] under the control of the $P_{BAD}$ promoter induced by L-arabinose in the plate media[36]. Rhamnolipid investment was tracked through the $P_{rhlAB}$-GFP promoter fusion[28,37]. The $P_{rhlAB}$-GFP construct has been shown not to phenocopy constitutive gene expression in the spatially-structured system[22]. When $rhlAB$ is driven by the $P_{BAD}$ promoter with the same w/v concentration as we use here (1.5%), the constitutive cooperator consistently loses to the defector strain $\Delta rhlA$ in competition[27]. We further found that in our data, DsRed and GFP fluorescence do not scale together (Supplementary Fig. 14).

To supplement the identification of colonies, we developed a method to separate colonies that grow together and "merge" over the course of the timeseries so they could be tracked independently. After the images were background corrected, the peaks of the colonies were identified across a range of images and parameter values. The images used were taken between 20 and 30 h, before the majority of colony merge events. The best parameters for peak identification were selected and used in the downstream analysis.

Each complete image timeseries was used to create a mask with all pixels that eventually contained biomass. Once identified, each pixel was tracked throughout the timeseries. To localize pixels to their cognate colony, the previously identified peaks, the mask and the biomass distribution in the final timepoint were used with the watershed algorithm to identify the boundaries of colony objects.

**Analysis of swarming tendrils**. Swarming tendrils were analyzed using the same process as the immotile colonies wherever possible. We used tendrils that did not branch to control for any variation that may arise due to the branching process.

To determine the speed of a moving tendril, the location of the edge was calculated every 5 min and the data were smoothed with a moving window of 25 min.

To isolate the time of tendril formation, swarms were imaged in a prototype imager equipped with a fish eye lens allowing for the acquisition of brightfield data for up to 12 swarming plates at a time. The timeseries were analyzed in ImageJ to identify the time of tendril formation. As the fish eye lens spreads the image pixels to cover a much larger region, the signal to noise ratio was managed carefully when collecting these data. Tendril formation times for each plate were calculated at three different zoom levels and averaged. To avoid bias, the data for each plate was collected by at least two independent researchers before averaging. The dataset for swarms without quorum signals includes eight biological replicates with 80 total technical replicates. The dataset for swarms with exogenously provided quorum signals includes six biological replicates with 54 total technical replicates.

Swarming competition results were collected by CFU assay. After each 24 h competition, the cells were washed off of the plates using PBS. The recovered cells were diluted and counted to determine the outcome of the competition[22,27].

**Independence of measurements in statistical comparisons**. In the analysis of Fig. 1e, we describe the rank sum results as $p$-value < 1e−4. This analysis is performed using points taken every 2 h iteratively from the timeseries for statistical analysis in order to control for timepoint independence. All $p$-values were <1e−4.

In Fig. 5c, we compare between swarming tendrils grown with and without exogenously provided quorum signals to determine if there is a difference in the distribution of specific growth rate in the tendril tip with this perturbation. We include all timepoints as time did not explain the variation in the growth rate data (tendrils without quorum signals, $R^2 = 0.02$, tendrils with quorum signals, $R^2 = 0.15$).

**Reporting summary**. Further information on research design is available in the Nature Research Reporting Summary linked to this article.

## Data availability
The data generated in this study, including all data shown in the figures, are available in the Supplementary Information and the Source Data file. The raw data can be found at https://figshare.com/projects/Spatial-temporal_microbial_cooperation/124954. Source data are provided with this paper.

## Code availability
Code for these analyses is available in the github repository at https://github.com/htlm/ImageTimeseriesAnalysis.

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

## Acknowledgements

The authors acknowledge Ned Wingreen, Chris Myers, Kyu Rhee, Dan Heller, Jinyuan Yan, and Chen Liao for helpful discussions and paper comments. This work was funded by National Science Foundation (www.nsf.gov) award MCB- 1517002/NSF 13-520 to J.B.X., as well as National Science Foundation Graduate Research Fellowships under Grant No. 1257284 to H.M. and Grant No. 1946429 C.C.R.

## Author contributions

H.M. and J.B.X. designed study. M.D. designed and constructed the fluorescent time-series imaging hardware and associated software. H.M. wrote the image timeseries data analysis software for data extraction, analyses and statistical inference model. H.M., K.S.L., T.S., Y.C. and C.C.R. performed experiments. H.M., K.S.L., T.S., Y.C. and C.C.R. analyzed data. H.M., B.P.T. and J.B.X. designed the statistical inference model. H.M. and J.B.X. wrote the paper. All authors provided feedback on the paper.

## Competing interests

The authors declare no competing interests.
