## [Peer Review File · Nature Communications]

Spatial-temporal dynamics of a microbial cooperative behavior resistant to cheatingReviewers' comments:

Reviewer #1 (Remarks to the Author):

The authors of this manuscript address how *P. aeruginosa* regulates cooperative behaviors in response to multiple inputs, namely quorum sensing and nutrient status, to prevent public goods cheating. The authors employ a reporter for the high-cell-density quorum-sensing state, PrhIAB-GFP, as a readout for cooperativity in swarming and non-swarming growth modes. To monitor fluorescence during growth on plates, the authors construct a home-built imaging setup that is the primary experimental tool used in the work. The authors find that the PrhIAB-GFP signal is heterogeneous when cells are grown on plates, in contrast to what has been previously observed in liquid cultures. In conditions that permit swarming, PrhIAB-GFP signal peaks at swarm edges. In colony growth conditions, PrhIAB-GFP output depends on the cell density of the local region. The authors use their data to generate various models to describe the behavior they observe. The authors conclude that the integration of multiple cues stabilizes cooperation.

While the authors address an interesting and relevant question, in its current state, the findings in the manuscript are very preliminary, and in several instances, the claims are not underpinned by data. In other places, the data clearly show something different from what the authors state. There are also issues with lack of controls in experiments. Thus, it is not yet possible to assess the validity of the authors' findings and their interpretations. There are many errors in grammar and word use, there are typos and logic gaps suggesting that the paper was hastily put together or, perhaps, an early rough draft was mistakenly submitted. A careful, full going over of the paper and significant rewriting is needed to address these issues. The comments provided below are not a comprehensive list of all issues, but rather, are intended to highlight a few examples of systemic problems in the manuscript. Hopefully, the authors will find these points helpful.

1. The authors must ensure that the observed heterogeneity in fluorescence signal on plates is not due to inherent properties of the fluorescent proteins (FPs) they use as reporters. It is well known that the photophysical properties of FPs change with local pH, oxygen concentrations, etc. Moreover, different FPs respond uniquely to discrete environmental conditions. To prove that such phenomena do not underly the observed signal heterogeneity, it is standard practice to exchange the FPs used for each reporter. In the context of this work, the authors need to generate and test PrhIAB-DsRed and PBad-GFP as a control for the work. These results can be presented in the supplemental information for swarming and colony conditions (as in Figure 1A and 4A) and will verify that the differences in fluorescence signal are consequences of differential gene expression from the rhIAB promoter.

2. In several instances, the interpretations provided by the authors do not match the results shown. One major example involves Figure 1D. From the text, beginning line 196:

"In the center and mid-tendrils of the swarm, we observe that high growth rates correlate with lower levels of per capita gene expression (Figure 1d dashed circle). This indicates that the cells may indeed be titrating gene expression in accordance with local nutrient availability as in liquid culture (Boyle et al. 2015). However, at the edge we find no correlation between rhIAB expression and measured growth rate."

First, there are no points within the dashed circle that represent the center of the swarm (red dots), thus the authors cannot claim a correlation between growth rates and lower levels of per capita gene expression in this region. Second, from this analysis, it appears that per capita gene expression of PrhIAB-GFP is in fact highest in the center of the swarm, the opposite of what was concluded earlier in the paper (i.e., in Figure 1B and a main take home point of the paper). Is something mislabeled? Finally, if the authors are going to discuss correlations, they must provide a correlation coefficient to back up their claims.

3. Figure 3C and its associated text are unclear. As presented, the plot is difficult to interpret. What is the "colony red" label on the x-axis of Figure 3C? What do the values on the x-axis represent? What does a high or low coefficient value indicate? The discussion in the text and legend are also very

confusing, and the text does not tell the reader what the panel is presenting nor what is the take home point.

4. In Figure 5 A-C, why are there no controls without autoinducer added for the “alternative configuration”? It is not legitimate to compare the results from this setup to the control from the standard configuration setup. A no autoinducer control in the alternative configuration experiment is essential to evaluate the data.

5. In the figures with petri plates and two colors, it would be helpful to provide the images from each fluorescence channel independently, in addition to the merged images, (especially Figures 1A and 3A). As presented, it is not possible to discern the signal intensity for each channel independently when they are overlaid. Does the DsRed signal change from the center to the edge of the swarm?

6. In all figures, the legends are overly long. The authors heavily interpret the results in the legends when the legends should describe what is being presented in the figures. The legends are very confusing as written.

7. In Line 221, the authors state that colony development was tracked from single cells to mature colonies. This is not accurate, as the colonies were below the detection limit until after ~18 hours of growth (at least this is where the authors show in the plots). This issue needs to be remedied.

8. Line 255, I believe the authors mean “thousands of measurements”, rather than “thousands of independent experiments”?

9. In Figure 2D, the relationship between colony biomass and colony density is not notable until late time points. These data would be better visualized by presenting only the results from the final timepoint, perhaps as a scatter plot.

10. In several cases, the authors state that rhlAB expression is “turned off” or “shut off” under conditions of nutrient depletion. That is not the case, a more appropriate statement would be “decreased” or “reduced” as signal does not decline to zero.

11. In Figure 4C, why does activation not occur in a roughly uniform radial fashion? It appears that, despite relatively similar colony cell densities, the top half of the plate has higher PrhlAB-GFP expression than the bottom half irrespective of the distance from the autoinducer. Something seems amiss here.

12. Line 510, it has recently been shown that surface attachment does modulate quorum sensing gene expression in *P. aeruginosa*. (Chuang et al. 2019, <https://doi.org/10.1038/s41467-019-12153-1>). Thus, this statement is incorrect/out of date.

13. It seems as if an early draft of the paper was mistakenly sent in. There are many (many) grammatical errors. The manuscript needs to be heavily edited. Here are a very few examples: “microbes living inside a densely microbial community,” and “the edge of the tendril as moves forward.” The word “queue” is used when the authors mean “cue”. The sentence, “...can be described using an ecological kernel...” makes no sense. “We focus on the gram negative *Pseudomonas aeruginosa*...”, they must mean gram-negative bacterium....There are too many errors to lay out here.

Reviewer #2 (Remarks to the Author):

The authors present an investigation of cooperator-cheat dynamics in *Pseudomonas aeruginosa*, focussing on the way these dynamics are influenced, spatially and temporally, during swarming. As the authors point out, most of the work investigating cooperation in bacteria relies on observation of cells behaviour in liquid culture, which is unlikely to relate to conditions in natural populations. The

paper has, therefore, the potential to make a valuable contribution to our understanding of how social behaviours evolve in scenarios they might typically encounter – how do these dynamics unfold at the tip of an expanding colony vs in the centre, and over what scale? There is some beautiful data in here and the data is very nicely presented in figures. The problem is, I find it difficult to identify fully-supported, specific insights that emerge from their results, and their interpretations rely on assumptions that are not, to my knowledge, supported by data.

To improve the manuscript and maximise its potential impact, I suggest the following clarifications would be helpful:

- (1) Line 22 and throughout: why is it “counterintuitive” for the expression of rhamnolipid, a surfactant required for motility, to peak at the swarming edges? Surely this is exactly where we’d expect it to be expressed?
- (2) Line 39 – proximate and ultimate causation is confused throughout - robustness of cooperative behaviours depend on whether or not they are favoured by selection, not “cell-level computation”.
- (3) Lines 90-98 – This is a passage of text that seems key to setting up the specific hypothesis being tested. I didn’t find it easy to follow: I think the point is that cells rely on a variety of cues about their social and abiotic environments in order to optimise their behaviour? But what is the specific outstanding question?
- (4) Lines 103-104 – the assumption that QS signals are weakest at the tips of an expanding colony is key to the authors’ interpretation of their results and isn’t supported by data. I suspect QS expression in a growing colony to be quite complex: many of the QS controlled excretions are likely to be required at the growing edge, so perhaps we’d expect expression patterns to reflect that. Distribution of QS signal underlies the main conclusion of the paper, that cells are “integrating cues”. I can’t see how this can be stated without knowing how QS signaling varies across the colony.
- (5) The controls on hard agar are a nice addition to the story, however, I wonder how to interpret the response of bacterial cells to these conditions if they have not been selected to respond optimally. Is it like putting a dog on the moon and trying to understand dogs better by watching how they respond to zero gravity?
- (6) Is the result that QS communication occurs over cm scale a novel finding?
- (7) The emphasis of the title suggests that this is a paper about the “robustness” of cooperative behavior to cheating but I didn’t get a clear idea of the contribution of the results to this specific issue.

In general, I hope the authors will think about what specific hypotheses they are testing here and how far their data takes them towards answering them. They have collected some really beautiful data that could form the basis of an excellent (shorter and tighter) paper.

Minor comments

Lines 44-49 – this section of the text lacks citations.

Line 52 – “ecological kernel” should be defined for the general reader, certainly for microbiologists who will make up a large proportion of readers!

Line 64 – what does “evolutionary robustness” mean here in relation to swarms?

Lines 73-74 – writing a bit sloppy here. Perhaps should read “..integrates quorum signals and information about nutrient availability to optimize timing of rhamnolipid secretion (Xavier...”

Line 90 – delete “According to the literature” and what literature? Only Cornforth is cited.

Line 195 – what is specifically meant by “metabolically prudent dynamics” here?

Lines 494-499 – text hard to follow here.

Lines 514-515 – I’m a bit lost again - what specific hypothesis was being tested by “no growth cost under social perturbation”?

Lines 519-522 – yes, exactly. But this reads like an afterthought or something added to appease a reviewer(!) Given these potential unknown complexities, the paper would benefit from a more measured and specific test of hypotheses.

We thank the two anonymous reviewers for their generosity in the time and thoughtful comments they provided for our manuscript. Below is a point-by-point breakdown describing how we have addressed the reviewer comments in our new manuscript.

Reviewers' comments:

Reviewer #1 (Remarks to the Author):

The authors of this manuscript address how *P. aeruginosa* regulates cooperative behaviors in response to multiple inputs, namely quorum sensing and nutrient status, to prevent public goods cheating. The authors employ a reporter for the high-cell-density quorum-sensing state, PrhIAB-GFP, as a readout for cooperativity in swarming and non-swarming growth modes. To monitor fluorescence during growth on plates, the authors construct a home-built imaging setup that is the primary experimental tool used in the work. The authors find that the PrhIAB-GFP signal is heterogeneous when cells are grown on plates, in contrast to what has been previously observed in liquid cultures. In conditions that permit swarming, PrhIAB-GFP signal peaks at swarm edges. In colony growth conditions, PrhIAB-GFP output depends on the cell density of the local region. The authors use their data to generate various models to describe the behavior they observe. The authors conclude that the integration of multiple cues stabilizes cooperation.

While the authors address an interesting and relevant question, in its current state, the findings in the manuscript are very preliminary, and in several instances, the claims are not underpinned by data. In other places, the data clearly show something different from what the authors state. There are also issues with lack of controls in experiments. Thus, it is not yet possible to assess the validity of the authors' findings and their interpretations. There are many errors in grammar and word use, there are typos and logic gaps suggesting that the paper was hastily put together or, perhaps, an early rough draft was mistakenly submitted. A careful, full going over of the paper and significant rewriting is needed to address these issues. The comments provided below are not a comprehensive list of all issues, but rather, are intended to highlight a few examples of systemic problems in the manuscript. Hopefully, the authors will find these points helpful.

We thank the reviewer for their rigorous feedback and have found their points significantly helpful to the improvement of the manuscript.

1. The authors must ensure that the observed heterogeneity in fluorescence signal on plates is not due to inherent properties of the fluorescent proteins (FPs) they use as reporters. It is well known that the photophysical properties of FPs change with local pH, oxygen concentrations, etc. Moreover, different FPs respond uniquely to discrete environmental conditions. To prove that such phenomena do not underly the observed signal heterogeneity, it is standard practice to exchange the FPs used for each reporter. In the context of this work, the authors need to generate and test P_{rhIAB} -DsRed and P_{BAD} -GFP as a control for the work. These results can be presented in the supplemental information for swarming and colony conditions (as in Figure 1A and 4A) and will verify that the differences in fluorescence signal are consequences of differential gene expression from the *rhIAB* promoter.

We thank the reviewer for this comment. In this request, the reviewer asks for verification that (1) the constitutively expressed DsRed via the P_{BAD} -DsRed construct is an accurate estimation of constitutive expression and thus reflective of biomass and (2) whether constitutively expressed GFP and GFP driven by P_{rhlAB} are truly different.

(1) We observe that our P_{BAD} -DsRed construct is constitutively produced, as evidenced by the positive correlation between fluorescence and cell number (Supp. Figure 1).

(2) We find that our results are indicative of gene expression differences between the constitutive P_{BAD} promoter and the differentially expressed P_{rhlAB} . It has previously been shown that the usage of these two promoters to control *rhlAB* gene expression leads to differing gene expression patterns in the spatial environment (de Vargas Roditi et al, 2013). In that paper, the authors showed that the difference between these expression patterns leads to systemic alterations in cooperative behavior and competitive outcome against a rhamnolipid defector. It has also previously been shown in the spatial context that the P_{rhlAB} -GFP construct does not phenocopy a constitutively expressed GFP (Xavier et al, 2011 (Figure 3c)). In addition, we have added side-by-side comparisons of the P_{BAD} -DsRed and P_{rhlAB} -GFP data in the swarms as Supp. Figure 8. For further comparison, the L-arabinose concentration used in driving *rhlAB* in de Vargas Roditi et al 2013 (1.5% w/v) is the same concentration we use to induce the DsRed in our data. Taken together, these data show that the differences we observe between the GFP and DsRed fluorescence signals are consequences of differential gene expression from the *rhlAB* promoter.

We regret that the reviewer's proposed color swap (P_{rhlAB} -DsRed and P_{BAD} -GFP), is not a feasible control for fluorescence signal in this system. The GFP marker cannot be used as a marker for biomass localization in spatial structure due to a fluorescent halo effect. However, the difference between constitutive GFP and P_{rhlAB} -GFP documented in Xavier et al, 2011, figure 3c shows that there is differential gene expression information in the fluorescence data between the P_{rhlAB} promoter and constitutive expression beyond any halo effect. We have added a statement describing our choices of fluorescent markers in greater detail in the methods text describing the analysis of the immotile colonies, lines 494 - 503.

2. In several instances, the interpretations provided by the authors do not match the results shown. One major example involves Figure 1D. From the text, beginning line 196: "In the center and mid-tendrils of the swarm, we observe that high growth rates correlate with lower levels of per capita gene expression (Figure 1d dashed circle). This indicates that the cells may indeed be titrating gene expression in accordance with local nutrient availability as in liquid culture (Boyle et al. 2015). However, at the edge we find no correlation between *rhlAB* expression and measured growth rate."

First, there are no points within the dashed circle that represent the center of the swarm (red dots), thus the authors cannot claim a correlation between growth rates and lower levels of per capita gene expression in this region. Second, from this analysis, it appears that per capita gene expression of P_{rhlAB} -GFP is in fact highest in the center of the swarm, the opposite of what was concluded earlier in the paper (i.e., in Figure 1B and a main take home point of the paper). Is something mislabeled? Finally, if the authors are going to discuss correlations, they must provide a correlation coefficient to back up their claims.

We thank the reviewer for noticing this. The legend in original figure 1d was in error and should have matched those in the original figures 1b and 1c.

We have replaced the per capita gene expression previously shown with a promoter activity investigation (Figure 4), which we believe showcase the data more appropriately. These data, in addition to Figure 1, more clearly showcase the differences between gene expression in the liquid and spatial (swarming) system.

3. Figure 3C and its associated text are unclear. As presented, the plot is difficult to interpret. What is the “colony red” label on the x-axis of Figure 3C? What do the values on the x-axis represent? What does a high or low coefficient value indicate? The discussion in the text and legend are also very confusing, and the text does not tell the reader what the panel is presenting nor what is the take home point.

We apologize for any confusion. After receiving this comment, we felt the presentation of the data was insufficient to showcase the spatial-temporal nature of colony-colony interactions we saw in our dataset. We have replaced this analysis with a data-driven model selection approach, see Figure 3. We hope this new presentation, now capturing and characterizing the length-scale of interaction between colonies, is clearer.

4. In Figure 5 A-C, why are there no controls without autoinducer added for the “alternative configuration”? It is not legitimate to compare the results from this setup to the control from the standard configuration setup. A no autoinducer control in the alternative configuration experiment is essential to evaluate the data.

In light of Figure 1 and to facilitate a clear message about the discrepancy between gene expression in the liquid and spatial environments, we have focused our analysis on a single comparison between colonies that were or were not provided quorum signals exogenously. This comparison is in Supplemental Figure 5. The data included is the same data included in Figure 1.

5. In the figures with petri plates and two colors, it would be helpful to provide the images from each fluorescence channel independently, in addition to the merged images, (especially Figures 1A and 3A). As presented, it is not possible to discern the signal intensity for each channel independently when they are overlaid. Does the DsRed signal change from the center to the edge of the swarm?

We have added these swarm images as Supplemental Figure 8 with further tendril-level detail in Supplemental Figures 10 and 11.

6. In all figures, the legends are overly long. The authors heavily interpret the results in the legends when the legends should describe what is being presented in the figures. The legends are very confusing as written.

We have shortened simplified the information in the legends to all figures.

7. In Line 221, the authors state that colony development was tracked from single cells to mature colonies. This is not accurate, as the colonies were below the detection limit until after ~18 hours of growth (at least this is where the authors show in the plots). This issue needs to be remedied.

The data were collected from the timepoint when the spotted cell suspension dried on the plate onwards. At this point the cells that formed each colony were undetected but were—by definition—CFUs. Only data from timepoints when the colony biomass came above detectable levels was shown. We have clarified this in our methods, lines 514-516.

8. Line 255, I believe the authors mean “thousands of measurements”, rather than “thousands of independent experiments”?

We have removed this line from the text.

9. In Figure 2D, the relationship between colony biomass and colony density is not notable until late time points. These data would be better visualized by presenting only the results from the final timepoint, perhaps as a scatter plot.

We show in Figure 1 that there is a positive correlation between colony promoter activity (P_{rhIAB}) and growth rate. As such, we believe the entirety of the growth curve and all growth states therein are important to show in the description of our results. We hope the layout of Figure 1 communicates this effectively.

10. In several cases, the authors state that rhIAB expression is “turned off” or “shut off” under conditions of nutrient depletion. That is not the case, a more appropriate statement would be “decreased” or “reduced” as signal does not decline to zero.

We have made text alterations to remove or clarify [Lines 158-161] this in accordance with published literature (Boyle et al, 2015).

11. In Figure 4C, why does activation not occur in a roughly uniform radial fashion? It appears that, despite relatively similar colony cell densities, the top half of the plate has higher PrhIAB-GFP expression than the bottom half irrespective of the distance from the autoinducer. Something seems amiss here.

We find minimal evidence of any radial bias to our data. To investigate this, we took each 45° increment around the plate and compared the distribution of the maximal promoter activity in that region to all other regions. If bias was present, the biased increment would show a statistically different distribution of maximal promoter activity when compared to any other region. These distributions were compared using the rank sum test. We find that the strongest signal of this type occurs in the 90-135° increment. However, this region is not statistically different from all other regions, there are fewer samples in this region and all samples are within 2.75cm of the quorum signal source, a region of high variability in our data. This analysis has been included in Supplemental Figure 6.

12. Line 510, it has recently been shown that surface attachment does modulate quorum sensing gene expression in *P. aeruginosa*. (Chuang et al. 2019, <https://doi.org/10.1038/s41467-019-12153-1>). Thus, this statement is incorrect/out of date.

Thank you for this reference. We have updated our text to reflect its contents [Lines 66-68, 117-120] and believe the manuscript is much improved with this content.

13. It seems as if an early draft of the paper was mistakenly sent in. There are many (many) grammatical errors. The manuscript needs to be heavily edited. Here are a very few examples: “microbes living inside a densely microbial community,” and “the edge of the tendril as moves forward.” The word “queue” is used when the authors mean “cue”. The sentence, “...can be described using an ecological kernel...” makes no sense. “We focus on the gram negative *Pseudomonas aeruginosa*...”, they must mean gram-negative bacterium....There are too many errors to lay out here.

We thank the reviewer for these grammatical notes. These examples have been edited where they still exist in the text. We have also expanded the description of our kernel approach in ways we hope increases clarity (Lines 252-272, Figure 3a, b).

Reviewer #2 (Remarks to the Author):

The authors present an investigation of cooperator-cheat dynamics in *Pseudomonas aeruginosa*, focussing on the way these dynamics are influenced, spatially and temporally, during swarming. As the authors point out, most of the work investigating cooperation in bacteria relies on observation of cells behaviour in liquid culture, which is unlikely to relate to conditions in natural populations. The paper has, therefore, the potential to make a valuable contribution to our understanding of how social behaviours evolve in scenarios they might typically encounter – how do these dynamics unfold at the tip of an expanding colony vs in the centre, and over what scale? There is some beautiful data in here and the data is very nicely presented in figures. The problem is, I find it difficult to identify fully-supported, specific insights that emerge from their results, and their interpretations rely on assumptions that are not, to my knowledge, supported by data.

We thank the reviewer for their insightful feedback and have found their points significantly helpful to the improvement of the manuscript.

To improve the manuscript and maximise its potential impact, I suggest the following clarifications would be helpful:

(1) Line 22 and throughout: why is it “counterintuitive” for the expression of rhamnolipid, a surfactant required for motility, to peak at the swarming edges? Surely this is exactly where we’d expect it to be expressed?

We have attempted to clarify the differences between the liquid and spatial datasets that we observe. We hope the reviewer finds the new abstract and new formulation of Figure 1 and particularly lines 156-163 a clearer description.

(2) Line 39 – proximate and ultimate causation is confused throughout - robustness of cooperative behaviours depend on whether or not they are favoured by selection, not “cell-level computation”.

We have adjusted the text to more clearly distinguish between proximate and ultimate causation, [Lines 41-43].

(3) Lines 90-98 – This is a passage of text that seems key to setting up the specific hypothesis being tested. I didn’t find it easy to follow: I think the point is that cells rely on a variety of cues about their social and abiotic environments in order to optimise their behaviour? But what is the specific outstanding question?

We have rewritten the introduction to the manuscript to more clearly describe the questions being interrogated in this study, culminating in the investigations we pose in lines 80-81.

(4) Lines 103-104 – the assumption that QS signals are weakest at the tips of an expanding colony is key to the authors’ interpretation of their results and isn’t supported by data. I suspect QS expression in a growing colony to be quite complex: many of the QS controlled excretions are likely to be required at the growing edge, so perhaps we’d expect expression patterns to reflect that. Distribution of QS signal underlies the main conclusion of the paper, that cells are “integrating cues”. I can’t see how this can be stated without knowing how QS signaling varies across the colony.

We have removed this language from the text.

(5) The controls on hard agar are a nice addition to the story, however, I wonder how to interpret the response of bacterial cells to these conditions if they have not been selected to respond optimally. Is it like putting a dog on the moon and trying to understand dogs better by watching how they respond to zero gravity?

We find that the hard agar CFU analyses create a better conceptual framework to interpret the gene expression we observed in swarming colonies than liquid culture analyses. We find the new analyses, particularly in Figures 1 and 4 speak well to this.

(6) Is the result that QS communication occurs over cm scale a novel finding?

Yes. Previous work has only shown that cell aggregates of ~5000 cells are able to communicate over micrometer distances (Darch et al, 2018). We have reworked the text to clarify this point.

(7) The emphasis of the title suggests that this is a paper about the “robustness” of cooperative behavior to cheating but I didn’t get a clear idea of the contribution of the results to this specific issue.

We have attempted to clarify our findings in the context of the preservation of this cooperative behavior in a competitive context both in the title and Figure 5.

In general, I hope the authors will think about what specific hypotheses they are testing here and how far their data takes them towards answering them. They have collected some really beautiful data that could form the basis of an excellent (shorter and tighter) paper.

We thank the reviewer for their compliment of our data and for their thoughtful comments. We have found their feedback highly insightful and constructive in revising the manuscript.

Minor comments

Lines 44-49 – this section of the text lacks citations.

References have been added in the equivalent text of the new manuscript [Lines 44-54].

Line 52 – “ecological kernel” should be defined for the general reader, certainly for microbiologists who will make up a large proportion of readers!

We have better defined the kernel infrastructure we use, detailed in lines 252-272, with references to Figure 3a, b.

Line 64 – what does “evolutionary robustness” mean here in relation to swarms?

We have removed this phrase.

Lines 73-74 – writing a bit sloppy here. Perhaps should read “..integrates quorum signals and information about nutrient availability to optimize timing of rhamnolipid secretion (Xavier...”

This language has been clarified in lines 59-64.

Line 90 – delete “According to the literature” and what literature? Only Cornforth is cited.

This is clarified in lines 51-54.

Line 195 – what is specifically meant by “metabolically prudent dynamics” here?

This phrase has been removed.

Lines 494-499 – text hard to follow here.

These lines have been revised into lines 403-411.

Lines 514-515 – I’m a bit lost again - what specific hypothesis was being tested by “no growth cost under social perturbation”?

This section of text has been removed.

Lines 519-522 – yes, exactly. But this reads like an afterthought or something added to appease a reviewer(!) Given these potential unknown complexities, the paper would benefit from a more measured and specific test of hypotheses.

We thank the reviewer for this comment. We hope our revisions more clearly describe the role of quorum signal perturbation in this study. Using quorum signals as a perturbation mechanism has been used extensively [Lines 140-141], just not in this spatial format. We argue that the use of quorum signal perturbation in our experiments is a way to show that a quorum signal input – known here to act with transcription factors LasR (Pearson et al, 1994) and RhIR (Ochsner et al, 1994, Ochsner and Reiser, 1995) – can be provided and will institute a response from long range (Figure 2). Further, we apply the same perturbation in the swarming context to determine whether there can be any phenotypic impact to a colony receiving such input. We find that indeed, a colony receiving exogenous quorum signal input is able to protect itself against invasion by a defector despite (or perhaps due to) several additional phenotypic changes to the swarms (Figure 5).

Reviewers' comments:

Reviewer #2 (Remarks to the Author):

The authors have significantly improved the paper and I feel my concerns about the original submission have been addressed. It is a much clearer read and tighter story with the specific hypotheses expressed much more clearly. And the nice results are still there..
Nice work, everyone!

Reviewer #3 (Remarks to the Author):

Review of Monaco et al.

In this revised version of Monaco et al., the authors made substantial modifications to the manuscript – they changed the presentation of results, the order and logic of the manuscript, and edited the text such that findings were more clearly laid out for the reader. Overall, the manuscript improved from the initial version. That said, there are still many overarching problems that make the work difficult to evaluate and there lacks a clear “take away” from the manuscript. I have outlined some of the problems below, though, as before, this is not an exhaustive list.

1. There are many cases where better language precision is needed. This is not a minor issue; it makes the manuscript very difficult to interpret. For example, in the abstract, line 25, the authors state “expression of rhamnolipid genes in spatially-structured colonies.” To me, this implies that the authors are evaluating gene expression in single colonies (at the sub-colony level) that have spatial structure or heterogeneity within. As another example the authors refer to their colony imaging assay as a “CFU assay” (line 125). This is not a CFU assay, in which a researcher performs dilutions to count colony numbers. This issue is propagated throughout the manuscript. As another example, on line 250, the authors say, “we define colony-colony interactions here loosely.” Perhaps then, “interaction” is not the word to (repeatedly) use since, as you correctly imply, you are not actually modeling interactions. As a final example, line 372, the authors state “Promoter activity in the tendril tips of these tips achieved higher growth rates...”. I do not understand this statement. How can a promoter have a growth rate?

2. The organization of the manuscript is complicated to follow making a central theme or message difficult to ascertain. Transitions between figures are abrupt and the logic is not laid out well. This was especially true in the transition between Figure 3 and Figure 4. There needs to be clear transition sentences or paragraphs that help the reader understand why the authors did what they did. Related, though not as critical, why is the imaging system described in the Introduction? I think, aside from a sentence in the introduction stating that you developed a new plate-imaging setup, the description of the imaging system (which could be more thorough), belongs at the start of the Results section.

3. The authors compare homogenous liquid growth to heterogenous “spatially structured” growth on agar plates. The authors neglect to acknowledge a major difference between these two arrangements (aside from spatial organization), and that is growth in liquid vs growth on a solid substrate. This, in and of itself, changes quorum sensing gene expression (and probably the expression of many other genes), as was the point of the Chuang et al. paper that the authors now cite. Wouldn't it be better to make comparisons of homogeneous vs heterogenous spatial structure within one growth condition? In this way, one would be certain they are addressing differences spatial structure as opposed to a solid vs liquid difference. Perhaps analyzing colonies that are grown isolated on a plate, vs a plate with multiple colonies would be a better strategy.

4. Though the authors reference previous papers suggesting that the PrhIAB-GFP reporter does report on rhIAB activity and not on some environmental condition, I still believe, for the purposes of validating the new imaging setup and assay, that the authors should be able to swap the two fluorophores and observe the same relationship between biomass and rhIAB output. This is an easy control that should quickly validate the authors findings. Claiming that the Pbad-GFP exhibits a “halo effect” (not sure what this means optically), does not provide any assurance to me. If the halo effect is

a problem for the Pbad-GFP strain, why is it not a problem for PrhIAB-GFP? The optics are the same.

5. Line 95, "...study this behavior directly in the more natural spatial environment." I do not think it is fair to call this a more natural environment.

6. What information is contained within the figure 1c inset? It is difficult to see.

7. Line 235 ""in liquid known" should be "in liquid is known."

8. Tenses change in places, for example, in methods, line 462 "plate has 20mL" should be "plate had 20mL."

REVIEWER COMMENTS

Reviewer #2 (Remarks to the Author):

The authors have significantly improved the paper and I feel my concerns about the original submission have been addressed. It is a much clearer read and tighter story with the specific hypotheses expressed much more clearly. And the nice results are still there..
Nice work, everyone!

Thank you! We thank the reviewer for their time and fantastic comments in both iterations of this review process. We are excited and pleased that this new version communicates our findings clearly.

Reviewer #3 (Remarks to the Author):

Review of Monaco et al.

In this revised version of Monaco et al., the authors made substantial modifications to the manuscript – they changed the presentation of results, the order and logic of the manuscript, and edited the text such that findings were more clearly laid out for the reader. Overall, the manuscript improved from the initial version. That said, there are still many overarching problems that make the work difficult to evaluate and there lacks a clear “take away” from the manuscript. I have outlined some of the problems below, though, as before, this is not an exhaustive list.

We thank the reviewer for these thoughtful comments. We believe addressing these has greatly improved our communication of the impact of this manuscript.

1. There are many cases where better language precision is needed. This is not a minor issue; it makes the manuscript very difficult to interpret. For example, in the abstract, line 25, the authors state “expression of rhamnolipid genes in spatially-structured colonies.” To me, this implies that the authors are evaluating gene expression in single colonies (at the sub-colony level) that have spatial structure or heterogeneity within. As another example the authors refer to their colony imaging assay as a “CFU assay” (line 125). This is not a CFU assay, in which a researcher performs dilutions to count colony numbers. This issue is propagated throughout the manuscript. As another example, on line 250, the authors say, “we define colony-colony interactions here loosely.” Perhaps then, “interaction” is not the word to (repeatedly) use since, as you correctly imply, you are not actually modeling interactions. As a final example, line 372, the authors state “Promoter activity in the tendrils tips of these tips achieved higher growth rates...”. I do not understand this statement. How can a promoter have a growth rate?

We thank the reviewer for suggesting these language precision improvements. We feel that addressing these has improved the clarity of our manuscript.

We have revised line 25 to better summarize our study.

While we do perform dilutions and do count the resultant colonies, our assays with colony forming units go beyond simply counting colonies, extracting more general insights about bacterial communication as a whole through the usage of easily-

generated CFUs. The utilization of this classic microbiology framework to greater purpose and insight is a key part of our manuscript's message [Lines 95-97]. We have adjusted the text to reflect our application of the classic CFU assay protocol rather than its traditional application of estimating the number of viable bacteria in a sample [Lines 139-142, 276-277]. Similarly, we have revised our application of the name CFU, designating the term cCFU to indicate colonies that arise from a viable colony forming unit [Lines 141, 201, 337, 341, 343, 409, 410, 475, 510]

We have revised our description of colony-colony interactions to more precisely describe the definition by statistical association that we use here. [Line 256].

We have clarified line 372 [Now line 383].

2. The organization of the manuscript is complicated to follow making a central theme or message difficult to ascertain. Transitions between figures are abrupt and the logic is not laid out well. This was especially true in the transition between Figure 3 and Figure 4. There needs to be clear transition sentences or paragraphs that help the reader understand why the authors did what they did. Related, though not as critical, why is the imaging system is described in the Introduction? I think, aside from a sentence in the introduction stating that you developed a new plate-imaging setup, the description of the imaging system (which could be more thorough), belongs at the start of the Results section.

We thank the reviewer for these suggestions to improve the flow of our manuscript and clarify our message. We have added a results section to detail our plate-imaging description [Lines 105-125]. We have also revised the transitions between figures 3 and 4 [Lines 325-333] and throughout the work [Lines 185-186, 233-235, 344-346, 389].

We have worked to clarify our central message in the main text. As stated in the revised introduction [Lines 91-102], the primary take-away message from our work is that bacterial aggregates are able to change their gene expression in response to diffusible cues sent by neighbors that are centimeters apart. This message however is built upon the backs of two other driving themes of this work: 1 – Novel technologies can allow us to observe longitudinal development of microbial social behaviors through slight adaptations of readily available classic microbiology techniques. 2 – There are aspects of gene expression critical to social behaviors that change depending on whether the cells are in liquid culture or spatial structure (Figure 1 and Figure 5).

3. The authors compare homogenous liquid growth to heterogenous "spatially structured" growth on agar plates. The authors neglect to acknowledge a major difference between these two arrangements (aside from spatial organization), and that is growth in liquid vs growth on a solid substrate. This, in and of itself, changes quorum sensing gene expression (and probably the expression of many other genes), as was the point of the Chuang et al. paper that the authors now cite. Wouldn't it be better to make comparisons of homogeneous vs heterogenous spatial structure within one growth condition? In this way, one would be certain they are addressing differences spatial structure as opposed to a solid vs liquid difference. Perhaps analyzing colonies that are grown isolated on a plate, vs a plate with multiple colonies would be

a better strategy.

We thank the reviewer for this comment and agree that population growth dynamics differ significantly between liquid culture and spatially-structured environments. We do acknowledge these differences, cataloguing them with high time-resolution in Figure 1b (liquid culture) and 1c (spatial structure) [Lines 143-149]. In Figure 1, we show that differences gene expression behavior between liquid culture (Figure 1c) and spatial structure (Figure 1e) on plates with less than 70 colonies can be explained in terms of growth rate of the focal colony in these two environments.

We agree with the reviewer that changes in growth between liquid and spatial structure can lead to differences in quorum signal production, as Chuang et al showed. However, we know very little about how the resulting downstream gene expression is impacted or how sensitive these systems are to their spatial-temporal environment. Based on work by our lab and others, we expected to find a complex interplay between cellular growth and quorum signals that could impact the production of *rhlAB* as well as cooperative swarming behavior. As such, we felt an unsupervised approach was necessary to keep the length-scales we uncovered in our analyses unbiased (Figure 3) and explicitly include growth rate as a key co-variate in our statistical modeling framework (Figure 3e) [Lines 286-290]. We have revised [Line 244-245] to clarify this.

The growth of single bacterial aggregates in isolation, while important, has been characterized by others in recent years (Cremer et al, Nature, 2019, Yan et al, PNAS, 2016 and Darch et al, PNAS, 2018). Our intent here was to further this knowledge and, given our novel imaging infrastructure, investigate length- and time-scales that had never been able to be assayed before. Here, the heterogeneity of cCFU seeding provided the means to achieve this because the application of the classic CFU protocol gave us a random sample of length-scales to investigate for associations between the behavior of neighboring colonies [Lines 276-279].

4. Though the authors reference previous papers suggesting that the PrhlAB-GFP reporter does report on rhlAB activity and not on some environmental condition, I still believe, for the purposes of validating the new imaging setup and assay, that the authors should be able to swap the two fluorophores and observe the same relationship between biomass and rhlAB output. This is an easy control that should quickly validate the authors findings. Claiming that the Pbad-GFP exhibits a “halo effect” (not sure what this means optically), does not provide any assurance to me. If the halo effect is a problem for the Pbad-GFP strain, why is it not a problem for PrhlAB-GFP? The optics are the same.

We thank the reviewer for this request for further validation of our PrhlAB-GFP reporter. We performed an exhaustive investigation to validate this promoter utilizing RT-qPCR [Supplemental Figure 5]. We found that in both liquid culture and spatial structure, our reporter does indeed report on *rhlAB* activity across a range of environmental conditions [Lines 120-122, 519-552].

The reason the halo effect is not a problem for the P_{rhlAB}-GFP in our data is that we only included GFP data in our analysis where the DsRed constitutive fluorescence indicated the presence of biomass [Lines 510-512].

5. Line 95, "...study this behavior directly in the more natural spatial environment." I do not think it is fair to call this a more natural environment.

We have removed this phrase.

6. What information is contained within the figure 1c inset? It is difficult to see.

We thank the reviewer for letting us know about the size issue. The inset provides an example plate image from our experiments to give readers a visual of the data. We have adjusted this inset for easier viewing.

7. Line 235 "'in liquid known" should be "in liquid is known."

We have fixed this in what is now line 242.

8. Tenses change in places, for example, in methods, line 462 "plate has 20mL" should be "plate had 20mL."

We have fixed this in what is now lines 469.

Reviewers' comments:

Reviewer #3 (Remarks to the Author):

I am satisfied with the revisions made by the authors and I believe the work is ready for publication.